# Design, Synthesis, and Biological Evaluation of Aromatic Amide-Substituted Benzimidazole-Derived Chalcones. The Effect of Upregulating *TP53* Protein Expression

**DOI:** 10.3390/molecules25051162

**Published:** 2020-03-05

**Authors:** Lintao Wu, Yuting Yang, Zhijun Wang, Xi Wu, Feng Su, Mengyao Li, Xiaobi Jing, Chun Han

**Affiliations:** 1Department of Chemistry, Changzhi University, Changzhi, Shanxi 046011, China; ltwu@live.cn (L.W.); czxywzj@163.com (Z.W.); wuxi040252768@163.com (X.W.); sufeng1788@126.com (F.S.); lmysummer@126.com (M.L.); 2University of Michigan Medical School, Ann Arbor, MI 48109, USA; yutyang@umich.edu; 3School of Chemistry and Chemical Engineering, Yangzhou University, Yangzhou, Jiangsu 225002, China

**Keywords:** aromatic amide-substituted, chalcones, antiproliferative activities, *TP53* protein, upregulating

## Abstract

A series of benzimidazole-derived chalcones containing aromatic amide substituent were designed and synthesized. All of the chalcone compounds were tested for their in vitro antitumor activity against human cancer cell lines (HCT116, HepG2, A549, and CRL-5908). The antiproliferative activity of compounds **3**, **6**, **9**, **14**, **15**, **16** against HCT116 cells was significantly better than that that of 5-Fluorouracil (IC50: 94.63 µM). The antitumor activity of these compounds showed obvious differences between the wild type HCT116 and mutant HCT116 (*TP53*^−/−^) cells. A preliminary mechanistic study suggested that these compounds act by upregulating the expression of *TP53* protein in tumor cells without inhibiting the MDM2-*TP53* interaction.

## 1. Introduction

Malignant neoplasms, or cancers, belong to a class of diseases associated with defects in the regulatory loop that controls the dynamic stability of the intracellular environment (including proliferation, differentiation, as well as cell migration and death) [1,2]. Although some progress has been made in the treatment of cancers in the past few decades, cancers still have one of the highest mortality rates worldwide.

With the advances in cancer research, scientists have found that the *TP53* gene is among the tumor suppressor genes that is most frequently associated with human tumors [3]. Its biological function is to maintain the stability of the genome by regulating cell cycle arrest and inducing apoptosis [4]. Mutations in the *TP53* gene occur in about 50% of human cancers. However, in non-mutated tumors, *TP53* is inactivated by its inhibitors, such as MDM2, which block its transcription and lead to its degradation [5]. It has been reported that when the *TP53* gene is activated, tumors may be completely cleared [6,7].

Since cancer-driver mutations like those of the *TP53* gene are specifically selected during tumor evolution, tumor cells are particularly sensitive to the increase in *TP53* expression. Recently, a study by Martins et al. on the recombination of *T**P53* in established tumors in mice have shown that *T**P53* is a highly potent inhibitor of tumor growth without causing further toxicity, which supports the activation of *TP53* expression as a cancer treatment strategy [8]. In recent years, great progress has been made in targeting the MDM2-*TP53* interaction to improve *TP53* expression, and a series of small-molecular inhibitors with good inhibitory effect on the MDM2 protein have been developed, such as Nutlins [9,10], Imidazol [11,12], Benzodiazepines [13,14], Spirooxindole [15,16], Isoquinolinones [17], Pyrrolidone [18,19].

In our previous work, benzimidazole group-containing chalcones were found to have good antitumor activity in vitro and in vivo [20]. Moreover, the experimental results also revealed that the antitumor mechanism of these compounds is mediated through inhibition of the interaction between *TP53* and MDM2 [20].

It has been reported that the key protein-binding surface of MDM2-*TP53* interaction is three hydrophobic cavities [21]. Therefore, in this study, an aromatic ring was added in the way of amide bond connection based on previous study, hoping to enhance the hydrophobicity of the compounds and improve the binding ability to MDM2 protein.

Therefore, in this study, based on the previous research, the structure of these chalcones was further modified, and a series of benzimidazole-derived chalcones containing aromatic substituent groups were designed and synthesized. Measurement of their in vitro anti-proliferation activity against several tumor cell lines revealed that they all have satisfactory anti-tumor activity. In addition, the structure-activity relationship was preliminarily evaluated. The mechanism validation experimental results showed that, with Nutlin-3a as a positive control, these compounds exerted their antitumor activity by upregulating the expression of *TP53* protein in tumor cells without inhibiting the MDM2-*TP53* interaction. These mechanism validation experimental results were further verified by coimmunoprecipitation analysis and cell cycle analysis results.

## 2. Results and Discussion

### 2.1. Chemistry

All the compounds were synthesized from commercially available o-phenylenediamine (Scheme 1). First, o-phenylenediamine was condensed with lactic acid to obtain the intermediate 2-hydroxyethyl benzimidazole, which was oxidized with an equal amount of chromium trioxide in acetic acid under reflux, to obtain the intermediate 2-acetylbenzimidazole. Next, using sodium hydroxide as the base, 2-acetylbenzimidazole and multiple aromatic aldehydes were further condensed in ethanol and then acidified to obtain benzimidazole-α, β-unsaturated ketones. The obtained chalcones were acetylated and condensed with various arylamines in *N*, *N*-dimethylformamide (DMF) to ultimately obtain the target compounds.

### 2.2. In Vitro Antiproliferative Activity and Structure-Activity Relationship

The in vitro antitumor activity of the 16 obtained target compounds was determined in human colon cancer cell line HCT116, human liver cancer cell line HepG2, and human lung cancer cell lines A549 and CRL-5908 using the MTT assay with 5-fluorouracil, paclitaxel, and Nutlin-3a as positive control drugs. The results, shown in Table 1, revealed that all the target compounds had a good inhibitory effect on these four cancer cell lines. In fact, most compounds showed higher inhibitory activity than the positive control drugs 5-fluorouracil and Nutlin-3a. Most compounds exhibited a stronger antiproliferative activity against HCT116 cells. Furthermore, in this experiment, compared with 5-fluorouracil, the antiproliferative activity of compound 6 against HCT116 cells was about 100-fold higher.

Regarding the structure-activity relationship, the antitumor activity of all 16 derivatives was meticulously analyzed. The results revealed that the antitumor activity of the derivatives substituted at the 3-position (**3**, **6**, **9**, **12**, **15**) of the aromatic ring was higher than that of the same derivatives substituted at the 2-position or 4-position or unsubstituted benzene ring. It should be noted that while derivative 10 had a strong inhibitory activity towards HCT116 cells, it showed low activity towards the other three tumor cell lines, indicating certain selectivity.

### 2.3. Cell Cycle Analysis

In order to gain a better insight into the antiproliferative mechanism of these compounds, compounds **6** and **9**, as well as the positive control drug Nutlin-3a, were selected for further evaluation based on the in vitro antiproliferative activity results obtained with these compounds Figure 1. Additionally, the changes in the cell cycle distribution of HCT116 cells were analyzed using flow cytometry. The results showed that compound **9** was able to block HCT116 cells in the G2/M phase in a dose-dependent manner, and this ability was more potent than Nutlin-3a at the same concentration. Meanwhile, compound **6** also exhibits a certain G2/M blocking effect at high concentration. In addition, we also found that compound **9** showed an obvious ability to induce apoptosis especially at low concentrations, which was superior to Nutlin-3a. Compound 6 also could induce apoptosis at high concentrations. Therefore, both compounds **6** and **9** may exert their antiproliferative activity against HCT116 cells through G2/M blocking and apoptosis induction, which were significantly higher than Nutlin-3a.

### 2.4. Western Blot Analysis

In order to investigate the mechanism of these compounds, we detected the protein expression in HCT116 cells by Western blot analysis. As is all established, Nutlin-3a can activate the TP53 pathway in cancer cells by inhibiting the interaction between TP53 and MDM2. Then we firstly detected the protein-protein interaction between TP53 and MDM2 in HCT116 cells by Co-Immunoreaction after treated with 25 µM of compounds **6** and **9**. It can be seen from Figure 2a that compounds **6** and **9** did not increase the expression of MDM2, suggesting that they had no effect on the interaction between MDM2 and TP53.

Figure 2b shows that the expression of TP53, as well as its downstream protein p21 were upregulated in a dose-dependent manner, whereas the protein level of cdc2 was decrease following the treatment with 5 µM and 25 µM of compounds **6** and **9** for 72 h.

Taken together, these results compellingly indicated that these compounds could activated TP53 pathway without inhibiting the MDM2-TP53 interaction.

### 2.5. In Vitro Antiproliferative Activity Against HCT116 (p53^−/−^)

It has been reported that chalcone derivatives display antitumor effects by acting on TP53 protein-related pathways. In this study, the antitumor mechanism of the prepared chalcones was preliminarily explored. In order to determine whether the in vitro antitumor activity of these compounds depends on TP53, the selectivity for tumor cell lines with different genotype of compounds **3**, **6**, **9**, **14**, and **15**, which have better antitumor activity against HCT116 cells, was evaluated against HCT116 (*TP53*^−/−^) cells Table 2. The results showed that the inhibitory activity or the half maximal inhibitory concentration (IC50) of these compounds against HCT116 (*TP53*^−/−^) cells was much lower than that against wild-type HCT116 cells, which confirms that the antitumor effect of the target compounds is TP53 dependent.

## 3. Experimental Section

### 3.1. Synthesis and Characterization

The proton nuclear magnetic resonance (^1^H NMR) and ^13^C NMR spectra were recorded using tetramethylsilane (TMS) as the internal standard in deuterated dimethyl sulfoxide (DMSO-d_6_) or deuterated chloroform (CDCl_3_), respectively, on a Bruker spectrometer (Bruker BioSpin GmbH, Rheinstetten, Germany) at 500 MHz. High-resolution mass spectrometry (HRMS) spectra were recorded on a Bruker compact™ QTOF system (Bruker Daltonik GmbH, Bremen, Germany). The melting points (m.p.) were determined using an SRS-OptiMelt Automated Melting Point System (Stanford Research Systems, Inc., Sunnyvale, CA, USA) without correction. Flash column chromatography was performed with a silica gel (200–300 mesh) purchased from Qingdao Haiyang Chemical Co. Ltd., (Qingdao, Shandong, China).

#### 3.1.1. General Procedure for all Derivatives

Compounds **b**–**e** were synthesized using our previously reported method [20]. 4-(1-benzimidazol-2-propene)benzoic acid (compound **e**, 1.50 g, 5.00 mmol), thionyl chloride (SOCl_2_, 18 mL) and *N*,*N*-dimethylformamide (DMF, 0.50 mL) were added into a 100-mL round-bottom flask for reflux. After conducting the reaction for 8 h, the reaction mixture was dried in a rotary evaporator to obtain a layer of a brownish red solid. The next reaction was carried out directly, but adding potassium carbonate (K_2_CO_3_) and DMF (18 mL). After the mixture was completely dissolved in an ice bath, the substituted aromatic amines were added and the reaction was continued in the ice bath. After conducting the reaction e for 4 h, the reaction mixture was dried in a rotary evaporator, and a dichloromethane (20 mL) and methanol (10 mL) were added. Extraction was then performed with distilled water. Then, the water phase was evaporated in the rotary evaporator to obtain the solid product (**1**–**16**).

#### 3.1.2. (*E*)-4-(3-(1*H*-benzo[d]imidazol-2-yl)-3-oxoprop-1-en-1-yl)-*N*-phenylbenzamide (**1**)

Following general procedure, compound **e** was reacted with aniline to afford compound **1** as a yellow solid (yield 55%), m.p. >300 °C. ^1^H NMR (500 MHz, CDCl_3_) δ9.62 (s, 1H), 8.02 (d, *J* = 7.5 Hz, 2H), 7.93 (d, *J* = 15.6 Hz, 2H), 7.67 (d, *J* = 7.5 Hz, 2H), 7.61–7.50 (m, 4H), 7.36–7.21 (m, 4H), 7.11–7.03 (m, 1H), 6.87 (d, *J* = 15.2 Hz, 1H).; ^13^C NMR (125 MHz, Common NMR Solvents) δ 179.61, 167.56, 143.42, 139.01, 138.79, 137.69, 137.50, 136.67, 135.28, 129.03, 128.77, 128.08, 126.05, 124.90, 123.62, 123.32, 122.48, 118.49, 115.12. HRMS (ESI) *m*/*z* calcd for [C_23_H_17_O_3_N_2_ + H]^+^, 368.1394; found, 368.1356.

#### 3.1.3. (*E*)-4-(3-(1*H*-benzo[d]imidazol-2-yl)-3-oxoprop-1-en-1-yl)-*N*-(o-tolyl)benzamide (**2**)

Following general procedure, compound **e** was reacted with o-toluidine to afford compound **2** as a yellow solid (yield 46%), m.p. >300 °C. 1H NMR (500 MHz, DMSO-d_6_) δ 9.32 (s, 1H), 8.12 (d, *J* = 15.2 Hz, 1H), 8.05 (d, *J* = 7.5 Hz, 2H), 7.90 (s, 1H), 7.77 (d, *J* = 7.5 Hz, 2H), 7.73–7.32 (m, 2H), 7.23 (dqd, *J* = 34.6, 7.4, 1.7 Hz, 4H), 7.12–7.03 (m, 3H), 2.24 (s, 3H).; ^13^C NMR (125 MHz, Common NMR Solvents) δ 179.61, 166.67, 143.42, 139.01, 138.79, 137.69, 137.50, 136.28, 135.28, 131.42, 129.83, 128.77, 128.08, 127.66, 126.22, 126.05, 123.62, 123.32, 118.49, 115.12, 17.35. HRMS (ESI) *m*/*z* calcd for [C_24_H_19_O_3_N_2_ + H]^+^, 382.1550; found, 382.1531.

#### 3.1.4. (*E*)-4-(3-(1*H*-benzo[d]imidazol-2-yl)-3-oxoprop-1-en-1-yl)-*N*-(m-tolyl)benzamide (**3**)

Following general procedure, compound **e** was reacted with m-toluidine to afford compound **3** as a white solid (yield 63%), m.p. 282.7–284 °C. 1H NMR (500 MHz, *d_6_*-DMSO) δ 9.72 (s, 1H), 8.02 (d, *J* = 7.5 Hz, 2H), 7.93 (d, *J* = 15.3 Hz, 2H), 7.67 (d, *J* = 7.5 Hz, 2H), 7.60–7.43 (m, 3H), 7.34–7.16 (m, 4H), 6.96 (dt, *J* = 7.5, 1.5 Hz, 1H), 6.87 (d, *J* = 15.2 Hz, 1H), 2.40 (s, 3H); 13C NMR (125 MHz, Common NMR Solvents) δ 179.61, 167.56, 143.42, 138.99, 138.79, 137.88, 137.69, 137.50, 135.28, 128.80, 128.08, 127.24, 126.05, 123.62, 123.32, 122.22, 120.66, 118.49, 115.12, 21.20. HRMS (ESI) *m*/*z* calcd for [C_24_H_19_O_3_N_2_ + H]^+^, 382.1550; found, 382.1513.

#### 3.1.5. (*E*)-4-(3-(1*H*-benzo[d]imidazol-2-yl)-3-oxoprop-1-en-1-yl)-*N*-(p-tolyl)benzamide (**4**)

Following general procedure, compound **e** was reacted with p-toluidine to afford compound **4** as a white solid (yield 59%), m.p. >300 °C. 1H NMR (500 MHz, *d_6_*-DMSO) δ 9.47 (s, 1H), 8.02 (d, *J* = 7.5 Hz, 2H), 7.93 (d, *J* = 15.6 Hz, 2H), 7.67 (d, *J* = 7.5 Hz, 2H), 7.60–7.48 (m, 2H), 7.3–.20 (m, 6H), 6.87 (d, *J* = 15.2 Hz, 1H), 2.38 (s, 3H).; ^13^C NMR (125 MHz, Common NMR Solvents) δ 179.61, 167.56, 143.42, 139.01, 138.79, 137.69, 137.50, 136.04, 135.28, 134.76, 129.33, 128.77, 128.08, 126.05, 123.62, 123.32, 120.53, 118.49, 115.12, 21.12. HRMS (ESI) *m*/*z* calcd for [C_24_H_19_O_3_N_2_ + H]^+^, 382.1550; found, 382.1497. 

#### 3.1.6. (*E*)-4-(3-(1*H*-benzo[d]imidazol-2-yl)-3-oxoprop-1-en-1-yl)-*N*-(2-chlorophenyl)benzamide (**5**)

Following general procedure, compound **e** was reacted with 2-chloroaniline to afford compound **5** as a bright yellow solid (yield 50%), m.p. 293.5–297.2 °C. 1H NMR (500 MHz, *d_6_*-DMSO) δ 9.60 (s, 1H), 8.10–7.95 (m, 2H), 7.92 (d, *J* = 6.9 Hz, 2H), 7.72 (s, 1H), 7.70–7.60 (m, 2H), 7.53 (t, *J* = 7.7 Hz, 3H), 7.37 (s, 1H), 7.26 (d, *J* = 10.9 Hz, 2H), 7.19 (s, 1H), 6.88 (s, 1H); ^13^C NMR (125 MHz, Common NMR Solvents) δ 179.61, 166.67, 143.42, 139.01, 138.79, 137.69, 137.50, 135.28, 134.89, 130.49, 128.77, 128.14, 127.85, 127.24, 126.06, 123.62, 123.32, 118.49, 115.12. HRMS (ESI) *m*/*z* calcd for [C_23_H_16_ClO_3_N_2_ + H]^+^, 402.1004; found, 402.0974.

#### 3.1.7. (*E*)-4-(3-(1*H*-benzo[d]imidazol-2-yl)-3-oxoprop-1-en-1-yl)-*N*-(3-chlorophenyl)benzamide (**6**)

Following general procedure, compound **e** was reacted with 3-chloroaniline to afford compound **6** as a light yellow solid (yield 67%), m.p. >300 °C. 1H NMR (500 MHz, *d_6_*-DMSO) δ 9.74 (s, 1H), 8.09–7.95 (m, 2H), 7.92 (d, *J* = 8.7 Hz, 2H), 7.71 (d, *J* = 2.2 Hz, 2H), 7.69–7.59 (m, 2H), 7.54 (d, *J* = 5.2 Hz, 2H), 7.35 (s, 1H), 7.26 (d, *J* = 11.1 Hz, 2H), 7.15 (s, 1H), 6.88 (s, 1H); ^13^C NMR (125 MHz, Common NMR Solvents) δ 179.61, 167.56, 143.42, 139.01, 138.72, 137.69, 137.50, 135.28, 133.77, 129.62, 128.77, 128.08, 126.05, 125.78, 123.62, 123.32, 121.90, 120.58, 118.49, 115.12. HRMS (ESI) *m*/*z* calcd for [C_23_H_16_ClO_3_N_2_ + H]^+^, 402.1004; found, 402.1050.

#### 3.1.8. (*E*)-4-(3-(1*H*-benzo[d]imidazol-2-yl)-3-oxoprop-1-en-1-yl)-*N*-(4-chlorophenyl)benzamide (**7**)

Following general procedure, compound **e** was reacted with 4-chloroaniline to afford compound **7** as a yellow solid (yield 60%), m.p. >300 °C. 1H NMR (500 MHz, *d_6_*-DMSO) δ 9.61 (s, 1H), 8.09–7.95 (m, 2H), 7.92 (d, *J* = 9.7 Hz, 2H), 7.74–7.59 (m, 2H), 7.56 (dd, *J* = 19.6, 2.7 Hz, 4H), 7.45–7.30 (m, 2H), 7.26 (d, *J* = 10.9 Hz, 2H), 6.87 (s, 1H); ^13^C NMR (125 MHz, Common NMR Solvents) δ 179.61, 167.56, 143.42, 139.01, 138.79, 137.69, 137.50, 136.20, 135.28, 129.97, 128.98, 128.77, 128.08, 126.05, 123.62, 123.32, 123.04, 118.49, 115.12. HRMS (ESI) *m*/*z* calcd for [C_23_H_16_ClO_3_N_2_ + H]^+^, 402.1004; found, 402.1100.

#### 3.1.9. (*E*)-4-(3-(1*H*-benzo[d]imidazol-2-yl)-3-oxoprop-1-en-1-yl)-*N*-(pyridin-2-yl)benzamide (**8**)

Following general procedure, compound **e** was reacted with 2-aminopyridine to afford compound **8** as a gray solid (yield 43%), m.p. >300 °C. ^1^H NMR (500 MHz, *d_6_*-DMSO) δ 8.39 (dd, *J* = 7.5, 1.4 Hz, 1H), 8.21 (d, *J* = 15.2 Hz, 1H), 8.03 (d, *J* = 7.5 Hz, 2H), 7.98–7.91 (m, 2H), 7.77 (d, *J* = 7.5 Hz, 2H), 7.71 (td, *J* = 7.4, 1.5 Hz, 1H), 7.58–7.51 (m, 2H), 7.35–7.22 (m, 3H), 6.88 (d, *J* = 15.0 Hz, 1H). ^13^C NMR (125 MHz, Common NMR Solvents) δ 179.61, 167.48, 152.17, 147.92, 143.42, 139.72, 139.01, 138.79, 137.69, 137.50, 135.28, 128.77, 128.08, 126.05, 123.62, 123.32, 119.04, 118.49, 115.88, 115.12. HRMS (ESI) *m*/*z* calcd for [C_22_H_16_O_4_N_2_ + H]^+^, 369.1346; found, 369.1317.

#### 3.1.10. (*E*)-4-(3-(1*H*-benzo[d]imidazol-2-yl)-3-oxoprop-1-en-1-yl)-*N*-(pyridin-3-yl)benzamide (**9**)

Following general procedure, compound **e** was reacted with 3-aminopyridine to afford compound **9** as a light gray solid (yield 40%), m.p. 204.6–206 °C. 1H NMR (500 MHz, *d_6_*-DMSO) δ 8.92 (d, *J* = 1.4 Hz, 1H), 8.30 (dd, *J* = 7.5, 1.3 Hz, 1H), 8.13 (d, *J* = 15.0 Hz, 1H), 8.03 (d, *J* = 7.5 Hz, 2H), 7.95 (s, 1H), 7.84 (d, *J* = 7.5 Hz, 1H), 7.77 (d, *J* = 7.5 Hz, 2H), 7.54 (dt, *J* = 6.9, 2.3 Hz, 2H), 7.33–7.22 (m, 3H), 7.07 (d, *J* = 15.0 Hz, 1H). ^13^C NMR (125 MHz, Common NMR Solvents) δ 179.61 (s, 1H), 167.56 (s, 2H), 147.61 (s, 2H), 143.38 (d, *J* = 10.5 Hz, 3H), 139.01, 138.79, 137.69, 137.50, 135.28, 133.22, 132.10, 128.77, 128.08, 126.05, 124.85, 123.62, 123.32, 118.49, 115.12. HRMS (ESI) *m*/*z* calcd for [C_22_H_16_O_4_N_2_ + H]^+^, 369.1346; found, 369.1327.

#### 3.1.11. (*E*)-4-(3-(1*H*-benzo[d]imidazol-2-yl)-3-oxoprop-1-en-1-yl)-*N*-(pyridin-4-yl)benzamide (**10**)

Following general procedure, compound **e** was reacted with 4-aminopyridine to afford compound **10** as a gray solid (yield 40%), m.p. 288.5–290.2 °C. 1H NMR (500 MHz, *d_6_*-DMSO) δ 8.50 (d, *J* = 7.4 Hz, 2H), 8.13 (d, *J* = 15.0 Hz, 1H), 8.05 (dd, *J* = 14.1, 7.4 Hz, 4H), 7.95 (s, 1H), 7.77 (d, *J* = 7.5 Hz, 2H), 7.54 (dt, *J* = 6.9, 2.3 Hz, 2H), 7.26 (pd, *J* = 7.5, 1.8 Hz, 2H), 7.07 (d, *J* = 15.2 Hz, 1H). ^13^C NMR (125 MHz, Common NMR Solvents) δ 179.61, 167.56, 153.12, 149.04, 143.42, 139.01, 138.79, 137.69, 137.50, 135.28, 128.77, 128.08, 126.05, 123.62, 123.32, 118.49, 115.12, 113.77. HRMS (ESI) *m*/*z* calcd for [C_22_H_16_O_4_N_2_ + H]^+^, 369.1346; found, 369.1317.

#### 3.1.12. (*E*)-4-(3-(1*H*-benzo[d]imidazol-2-yl)-3-oxoprop-1-en-1-yl)-*N*-(2-methoxyphenyl)benzamide (**11**)

Following general procedure, compound **e** was reacted with 2-methoxyaniline to afford compound **11** as a light yellow solid (yield 55%), m.p. 272.2–273.4 °C. ^1^H NMR (500 MHz, *d_6_*-DMSO) δ 8.84 (s, 1H), 8.02 (d, *J* = 7.5 Hz, 2H), 7.92 (t, *J* = 7.5 Hz, 2H), 7.72 (dd, *J* = 7.5, 1.4 Hz, 1H), 7.67 (d, *J* = 7.5 Hz, 2H), 7.54 (ddd, *J* = 6.7, 4.5, 2.1 Hz, 2H), 7.26 (pd, *J* = 7.5, 1.9 Hz, 2H), 7.12 (td, *J* = 7.4, 1.4 Hz, 1H), 7.02 (dd, *J* = 7.5, 1.5 Hz, 1H), 6.95 (td, *J* = 7.5, 1.5 Hz, 1H), 6.87 (d, *J* = 15.0 Hz, 1H), 3.94 (s, 3H). ^13^C NMR (125 MHz, Common NMR Solvents) δ 179.61, 166.67, 149.93, 143.42, 139.01, 138.79, 137.69, 137.50, 135.28, 128.77, 128.26, 128.08, 126.05, 123.62, 123.32, 122.80, 121.32, 118.49, 115.12, 112.37, 56.78. HRMS (ESI) *m*/*z* calcd for [C_24_H_19_O_3_N_3_ + H]^+^, 398.1499; found, 398.1464.

#### 3.1.13. (*E*)-4-(3-(1*H*-benzo[d]imidazol-2-yl)-3-oxoprop-1-en-1-yl)-*N*-(3-methoxyphenyl)benzamide (**12**)

Following general procedure, compound **e** was reacted with 3-methoxyaniline to afford compound **12** as a yellow solid (yield 59%), m.p. 281.7–283.6 °C ^1^H NMR (500 MHz, *d_6_*-DMSO) δ 9.69 (s, 1H), 8.02 (d, *J* = 7.5 Hz, 2H), 7.93 (d, *J* = 15.3 Hz, 2H), 7.67 (d, *J* = 7.5 Hz, 2H), 7.54 (ddd, *J* = 7.1, 5.2, 2.1 Hz, 2H), 7.42 (dt, *J* = 7.5, 1.4 Hz, 1H), 7.26 (pd, *J* = 7.4, 1.9 Hz, 2H), 7.19 (t, *J* = 7.5 Hz, 1H), 6.99 (t, *J* = 1.5 Hz, 1H), 6.87 (d, *J* = 15.2 Hz, 1H), 6.67 (dt, *J* = 7.5, 1.4 Hz, 1H), 3.83 (s, 3H). ^13^C NMR (125 MHz, Common NMR Solvents) δ 179.61, 167.56, 160.39, 143.42, 138.96, 138.79, 137.69, 137.50, 135.28, 129.20, 128.77, 128.08, 126.05, 123.62, 123.32, 118.49, 115.20, 110.44, 106.60, 56.03. HRMS (ESI) *m*/*z* calcd for [C_24_H_19_O_3_N_3_ + H]^+^, 398.1499; found, 398.1479.

#### 3.1.14. (*E*)-4-(3-(1*H*-benzo[d]imidazol-2-yl)-3-oxoprop-1-en-1-yl)-*N*-(4-methoxyphenyl)benzamide (**13**)

Following general procedure, compound **e** was reacted with 4-methoxyaniline to afford compound **13** as a light yellow solid (yield 70%), m.p. 258.4–260.2 °C. ^1^H NMR (500 MHz, *d_6_*-DMSO) δ 8.42 (s, 1H), 8.02 (d, *J* = 7.5 Hz, 2H), 7.93 (d, *J* = 15.9 Hz, 2H), 7.67 (d, *J* = 7.5 Hz, 2H), 7.57–7.46 (m, 4H), 7.26 (pd, *J* = 7.4, 1.9 Hz, 2H), 6.89 (dd, *J* = 14.5, 11.4 Hz, 3H), 3.87 (s, 3H). ^13^C NMR (125 MHz, Common NMR Solvents) δ 179.61, 167.56, 157.02, 143.42, 139.01, 138.79, 137.69, 137.50, 135.28, 130.94, 128.77, 128.08, 126.05, 123.62, 123.32, 122.99, 118.49, 115.12, 114.47, 56.03. HRMS (ESI) *m*/*z* calcd for [C_24_H_19_O_3_N_3_ + H]^+^, 398.1499; found, 398.1872.

#### 3.1.15. (*E*)-4-(3-(1*H*-benzo[d]imidazol-2-yl)-3-oxoprop-1-en-1-yl)-*N*-(2-hydroxyphenyl)benzamide (**14**)

Following general procedure, compound **e** was reacted with 2-aminophenol to afford compound **14** as a dark gray solid (yield 53%), m.p. >300 °C. 1H NMR (500 MHz, *d_6_*-DMSO) δ 9.25 (s, 1H), 8.62 (s, 1H), 8.17 (d, *J* = 15.2 Hz, 1H), 8.04 (d, *J* = 7.5 Hz, 2H), 7.77 (d, *J* = 7.5 Hz, 2H), 7.67 (dd, *J* = 7.8, 1.3 Hz, 1H), 7.56 (dd, *J* = 7.3, 1.6 Hz, 1H), 7.44 (dd, *J* = 7.3, 1.6 Hz, 1H), 7.26 (dddd, *J* = 12.0, 8.9, 7.4, 1.5 Hz, 3H), 6.95 (ddd*, J* = 7.4, 3.8, 2.5 Hz, 2H), 6.88 (d, *J* = 15.0 Hz, 1H), 6.16 (s, 1H). ^13^C NMR (125 MHz, Common NMR Solvents) δ 179.61, 166.67, 149.35, 143.42, 139.01, 138.79, 137.69, 137.50, 135.28, 128.77, 128.08, 127.37, 126.97, 126.05, 124.11, 123.62, 123.32, 120.78, 118.49, 115.86, 115.12. HRMS (ESI) *m*/*z* calcd for [C_23_H_17_O_3_N_3_ + H]^+^, 384.1343; found, 384.1351.

#### 3.1.16. (*E*)-4-(3-(1*H*-benzo[d]imidazol-2-yl)-3-oxoprop-1-en-1-yl)-*N*-(3-hydroxyphenyl)benzamide (**15**)

Following general procedure, compound **e** was reacted with 3-aminophenol to afford compound **15** as a gray solid (yield 62%), m.p. >300 °C. ^1^H NMR (500 MHz, *d_6_*-DMSO) δ 9.71 (s, 1H), 8.13 (d, *J* = 15.0 Hz, 1H), 8.04 (d, *J* = 7.5 Hz, 2H), 7.95 (s, 1H), 7.77 (d, *J* = 7.5 Hz, 2H), 7.56 – 7.52 (m, 2H), 7.26 (ddd, *J* = 13.6, 8.1, 6.3 Hz, 3H), 7.17 (t, *J* = 7.5 Hz, 1H), 7.08 (dd, *J* = 11.3, 8.0 Hz, 2H), 6.77 (dt, *J* = 7.5, 1.5 Hz, 1H), 5.80 (s, 1H). ^13^C NMR (125 MHz, Common NMR Solvents) δ 179.61, 167.56, 158.31, 143.42, 139.01, 138.72, 137.69, 137.50, 135.28, 130.62, 128.77, 128.08, 126.05, 123.62, 123.32, 118.49, 115.12, 113.52, 112.88, 108.15. HRMS (ESI) m/z calcd for [C_23_H_17_O_3_N_3_ + H]^+^, 384.1343; found, 384.1034.

#### 3.1.17. (*E*)-4-(3-(1*H*-benzo[d]imidazol-2-yl)-3-oxoprop-1-en-1-yl)-*N*-(4-hydroxyphenyl)benzamide (**16**)

Following general procedure, compound **e** was reacted with 4-aminophenol to afford compound **16** as a gray solid (yield 51%), m.p. >300 °C. ^1^H NMR (500 MHz, *d_6_*-DMSO) δ 8.42 (s, 1H), 8.13 (d, *J* = 15.2 Hz, 1H), 8.04 (d, *J* = 7.5 Hz, 2H), 7.95 (s, 1H), 7.77 (d, *J* = 7.5 Hz, 2H), 7.57–7.51 (m, 2H), 7.26 (pd, *J* = 7.5, 1.8 Hz, 2H), 7.08 (dd, *J* = 15.8, 11.3 Hz, 3H), 6.80 (d, *J* = 7.5 Hz, 2H), 5.60 (s, 1H). ^13^C NMR (125 MHz, Common NMR Solvents) δ 179.61, 167.56, 156.56, 143.42, 139.01, 138.79, 137.69, 137.50, 135.28, 129.97, 128.77, 128.08, 126.05, 124.38, 123.62, 123.32, 118.49, 116.04, 115.12. HRMS (ESI) *m*/*z* calcd for [C_23_H_17_O_3_N_3_ + H]^+^, 384.1343; found, 384.1372.

### 3.2. MTT Assay

The cellular growth inhibitory activities of the compounds were determined with MTT assays and A549, CRL-5908, HCT116 (+/+), HCT116 (−/−), and HepG2 cell lines (purchased from the American Type Culture Collection, (Gaithersburg, MD, USA). The cells were seeded at 5000 (100 µL)/well into a 96-well plate. After culturing in growth media at 37 °C for 24 h, the test compound dissolved in 100 mL of DMSO was added. The 5-Fuorouracil, Nutlin-3a, and Paclitaxel were used as the positive control (purchased from MedChem express (Monmouth Junction, NJ, USA). After 48 h of incubation, 20 mL of MTT solution (5 mg/mL) was added to each well, and the plates were incubated for 4 h at 37 °C with 5% CO_2_. After removing the culture medium, 150 mL of DMSO was added. The concentration of the compound that inhibited cell growth by 50% (IC_50_) was calculated. The assays were repeated in triplicate.

### 3.3. Cell Cycle Analysis

HCT116 cells (wild type p53) were seeded at 2 × 10^5^ cells/well in 6-well plates and cultured for 48 h. Next, the cells were incubated with the test compounds for 48 h. The cells were then treated with cold PBS (phosphate buffer saline). After harvest, the cells were fixed in 70% ice-cold ethanol overnight. Subsequently, the cells were centrifuged (1200 rpm for 5 min), the supernatant was discarded and the pellet was treated with RNase A (100 mg/mL) for 30 min at room temperature. The cells were stained using propidium iodide at a final concentration of 50 mg/mL. The stained cells were then analysed for cell cycle distribution using flow cytometry (BECKMAN, Atlanta, GA, USA), and the changes in the cell cycle profiles was analysed using (CELL QUEST PRO, Franklin lake, NJ, USA)

### 3.4. Western Blotting

HCT116 cells (wild type p53) were treated with various concentrations of the indicated compound for 48 h. Next, the cells were harvested by centrifugation at 1000 g for 5 min. The cell pellets were washed with PBS, resuspended in lysis buffer (150 mM NaCl, 50 mM Tris (pH 8.0), 0.02% NaN_3_, 0.01% PMSF, 0.2% Aprotinin, and 1% TritonX-100, supplemented with protease inhibitor cocktail (Thermo Scientific, Waltham, MA, USA), and centrifuged at 12,000 g for 10 min. The total protein concentration was determined using the Bio-Rad protein assay. The proteins were resolved by SDS-PAGE and transferred to a polyvinylidene fluoride membrane. After blocking with 5% non-fat milk in blocking buffer (PBS containing 0.1% Tween 20, pH 7.5), the membrane was incubated with the indicated primary antibody for 2 h at room temperature and then incubated with the appropriate peroxidase-conjugated secondary antibody. The immunoreactive bands were visualised using the ECL^TM^ Plus Western Blotting Detection System (Piscataway, New Jersey, USA). Beta-actin was used as a loading control.

## 4. Conclusions

In this study, a series of α, β-unsaturated ketone derivatives containing a benzimidazole group and an aromatic amide substituent were synthesized, and their biological activity was evaluated. All the compounds had strong antiproliferative activity against A549, HepG2, CRL-5908, and HCT116 cells. Among them, compounds **6** and **9** exhibited the strongest antiproliferative activity, which was much stronger than that of the positive control drugs, such as 5-fluorouracil, and Nutlin-3a. The preliminary structure-activity relationship analysis revealed that the substituent groups at the 3-position of the aromatic amides had a prominent effect on the activity of the compounds.

In order to further elucidate the antiproliferative mechanism of these compounds, the above two compounds were evaluated in a series of mechanism validation experiments. The results of the cell cycle analysis showed that these compounds, like the positive control drug Nutlin-3a, were able to block HCT116 cells in the G2/M phase and induce apoptosis in a dose-dependent manner. Western blot analysis results showed that to achieve their antiproliferative activity, these compounds significantly upregulated the expression of TP53 protein without inhibiting the MDM2 protein. The activity of these compounds against HCT116 (TP53^−/−^) cells was much weaker than that against wild-type HCT116 cells. These compounds were designed on the basis of our previous work, but their antitumor mechanism are quite different. This result should be related to the addition of aromatic amide-substituted group.

This study achieved a very interesting result, which changed our future research direction. In the future, we will further investigate the interactions between these compounds and p53, so as to further clarify the specific mechanisms of these compounds, and lay the foundation for exploring a class of leading compounds with clear mechanism and high anti-tumor activity.

In summary, these α, β-unsaturated ketone derivatives containing a benzimidazole group and an aromatic amide substituent can clearly upregulate the expression of TP53 protein and have excellent antiproliferative activity. They represent lead structures of novel antitumor drugs with great potential.

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
