# Peer review of "Design, Synthesis, and Biological Evaluation of Aromatic Amide-Substituted Benzimidazole-Derived Chalcones. The Effect of Upregulating TP53 Protein Expression"

_molecules, 2020, doi:10.3390/molecules25051162_

Round 1

Reviewer 1 Report

Minor Points: please provide the structures in Scheme 1 according to the ACS-1996 Style. "Figure 2 shows that the expression of TP53" and not Figure ().

Author Response

Point 1: please provide the structures in Scheme 1 according to the ACS-1996 Style. "Figure 2 shows that the expression of TP53" and not Figure (). 

Response 1: I modified the format of Scheme1 according to the ACS-1996 Style, and changed the statement about “Figure (2)” in this article

Reviewer 2 Report

In this study, benzimidazole-derived chalcones containing aromatic substituent groups were designed, synthesized and further evaluated for their antitumor
activity against human cancer cell lines. A mechanistic approach has also been incorporated to explore their upregulation through TP53 protein. 

This is an interesting and well-written work. Somehow I would suggest some minor corrections:

  1. The quality of Figure 1 is not suitable so please improve it. 
  2. The authors have not justified/explained anything about the necessity of modification of previously reported molecules. 
  3. I think possible applications of the work or future perspective should be considered. 

Author Response

Point 1: The quality of Figure 1 is not suitable so please improve it.

Response 1: I redownloaded the image from the flow cytometry and replaced it. Thank you very much

Point 2: The authors have not justified/explained anything about the necessity of modification of previously reported molecules.

Response 2: I have added this sentence in the Introduction: “It has been reported that the key protein-binding surface of MDM2-p53 interaction is three hydrophobic cavities. Therefore, in this study, an aromatic ring was added in the way of amide bond connection based on previous study, hope to enhance the hydrophobicity of the compounds and improve the binding ability to MDM2 protein.” Moreover, corresponding references have also been added.

Point 3: I think possible applications of the work or future perspective should be considered.

Response 3: I have added this sentence to the Conclusion: “This study achieved a very interesting result, which changed our future research direction. In the future, we will further investigate the interactions between these compounds and p53, so as to further clarify the specific mechanisms of these compounds, and lay the foundation for exploring a class of leading compounds with clear mechanism and high anti-tumour activity.”

Reviewer 3 Report

This paper reports the synthesis of a series of benzimidazole group-containing chalcones: the obtained compounds have been  evaluated for their in vitro antitumor activity against human cancer cell lines (HCT116, HepG2, A549, and CRL-5908). The novelty of this work with respect to a previous paper, reported by the same authors, lies on the insertion of an aromatic amide group as substituent: the synthetic approach exploits well known chemical processes. 

The summary affirms that the synthesized compounds show a moderate antiproliferative activity on the utilized cell lines, while the text  reports that all the target compounds had a good inhibitory effect on the examined cancer cell lines, showing higher inhibitory activity than the positive control drugs 5-fluorouracil and Nutlin-3a. I think that the authors must clarify this disagreement.

The suggested mechanism of action for the reported compounds does not appear very convincing. The authors need to a better and adequate support for their hypothesis.   Fig. 1 and Fig.2 need of a better explanation.

I think that the paper must be thoroughly revised and supported by more convincing biological data. 

Author Response

Point 1: The summary affirms that the synthesized compounds show a moderate antiproliferative activity on the utilized cell lines, while the text reports that all the target compounds had a good inhibitory effect on the examined cancer cell lines, showing higher inhibitory activity than the positive control drugs 5-fluorouracil and Nutlin-3a. I think that the authors must clarify this disagreement

Response 1: I am very sorry for the confusion that I have caused you. These compounds were obtained by further modification based on my previous work. Therefore, the moderate activity mentioned in the Abstract referred to the results of comparing the activity of these compounds with those reported previously. The higher activity mentioned in the text was the conclusion of comparing with positive drugs after the experiment, and the comparison objects were different. According to your suggestion, to avoid misunderstanding, I deleted the relevant description from the abstract.

Point 2: The suggested mechanism of action for the reported compounds does not appear very convincing. The authors need to a better and adequate support for their hypothesis.   Fig. 1 and Fig.2 need of a better explanation.

Response 2: Thank you for your advice. I have revised these two parts and added some explanations. Please check it.

2.3. Cell cycle analysis

In order to gain a better insight into the antiproliferative mechanism of these compounds, compounds 6 and 9, as well as the positive control drug Nutlin-3a, were selected for further evaluation based on the in vitro antiproliferative activity results obtained with these compounds. Additionally, the changes in the cell cycle distribution of HCT116 cells were analyzed using flow cytometry. The results showed that compound 9 was able to block HCT116 cells in the G2/M phase in a dose-dependent manner, and this ability was more potent than Nutlin-3a at the same concentration. Meanwhile, compound 6 also exhibits a certain G2/M blocking effect at high concentration. In addition, we also found that compound 9 showed an obvious ability to induce apoptosis especially at low concentrations, which was superior to Nutlin-3a. Compound 6 also could induce apoptosis at high concentrations. Therefore, both compounds 6 and 9 may exert their antiproliferative activity against HCT116 cells through G2/M blocking and apoptosis induction, which were significantly higher than Nutlin-3a.

2.4. Western blot analysis

In order to investigate the mechanism of these compounds, we detected the protein expression in HCT116 cells by Western blot analysis. As is all established, Nutlin-3a can activate the TP53 pathway in cancer cells by inhibiting the interaction between TP53 and MDM2. Then we firstly detected the protein-protein interaction between TP53 and MDM2 in HCT116 cells by Co-Immunoreaction after treated with 25 µM of compounds 6 and 9. It can be seen from Figure (2a) that compounds 6 and 9 did not increase the expression of MDM2, suggesting that they had no effect on the interaction between MDM2 and TP53.

Figure (2b) shows that the expression of TP53, as well as its downstream protein p21 were upregulated in a dose-dependent manner, whereas the protein level of cdc2 was decrease following the treatment with 5 µM and 25 µM of compounds 6 and 9 for 72 h.  

Taken together, these results compellingly indicated that these compounds could activated TP53 pathway without inhibiting the MDM2-TP53 interaction.

Round 2

Reviewer 3 Report

The authors have clarified all my observations. the manuscript has been significantly improved and ay my opinion warrants publication in Molecules.